# Dimensions of Online Conflict: Towards Modeling Agonism

**Matt Canute**[1], **Mali Jin**[2], **hannah holtzclaw**[1], **Alberto Lusoli**[1], **Philippa R Adams**[1], **Mugdha Pandya**[2], **Maite Taboada**[1], **Diana Maynard**[2] and **Wendy Hui Kyong Chun**[1]

[1]Digital Democracies Institute, Simon Fraser University, Canada, matthew_canute@sfu.ca
[2]Department of Computer Science, University of Sheffield, UK, m.jin@sheffield.ac.uk

## Abstract

Agonism plays a vital role in democratic dialogue by fostering diverse perspectives and robust discussions. Within the realm of online conflict there is another type: hateful antagonism, which undermines constructive dialogue. Detecting conflict online is central to platform moderation and monetization. It is also vital for democratic dialogue, but only when it takes the form of agonism. To model these two types of conflict, we collected Twitter conversations related to trending controversial topics. We introduce a comprehensive annotation schema for labelling different dimensions of conflict in the conversations, such as the source of conflict, the target, and the rhetorical strategies deployed. Using this schema, we annotated approximately 4,000 conversations with multiple labels. We then trained both logistic regression and transformer-based models on the dataset, incorporating context from the conversation, including the number of participants and the structure of the interactions. Results show that contextual labels are helpful in identifying conflict and make the models robust to variations in topic. Our research contributes a conceptualization of different dimensions of conflict, a richly annotated dataset, and promising results that can contribute to content moderation.

## 1 Introduction

Conflict is everywhere online. From political protests to spirited debates over the latest TikTok trend, these conflicts are simultaneously celebrated as promoting democracy and condemned for fostering polarization and undermining public institutions. Conflicts—ideas, arguments, or attitudes that oppose each other—are also central to platform moderation and monetization: from the amplification of certain controversies to provoke user engagement, to content takedowns to comply with a platform's terms of use or national laws (Gillespie, 2022, 2018; Morrow et al., 2022; Douek, 2022; Zeng and Kaye, 2022).

Not all conflicts are equal, and the confusion over the political and social value of conflict partly stems from its diverse nature. Conflict exists on a sliding scale ranging from antagonistic conflict between enemies (which is often silencing, undemocratic, and hateful because it is focused on delegitimizing opponents' rights and status) to agonistic conflict between adversaries (which has productive potential for the emergence of democratic dialogue, dissent, and trust in the public and political sphere, since the struggle is over interpretation and not legitimacy to speak) (Mouffe, 2002; Wenman, 2013). As many across the political spectrum have noted, democracy needs agonism to widen voices and prevent public spaces from becoming totalitarian or meaningless consensus hubs. But how do we know which conflicts are agonistic and which hateful? How do we know what kinds of counterspeech effectively facilitate dialogue and which destroy it? Which follow the spirit of platforms' terms of use and which do not?

To begin answering these questions, we present an in-depth exploration of conflict online, using the platform once known as Twitter (now "X") as a case study. Our work builds upon related Natural Language Processing (NLP) research fields such as abusive language, persuasion, and constructive comments. We use Twitter because, unlike the controlled environment of a priori conflictual discussions like those in Reddit's 'Change My View' (Monti et al., 2022; Srinivasan et al., 2019), it offers a more organic setting.

To determine the nature of a conflict, context is central (Zosa et al., 2021; Hu et al., 2022; Ghosh et al., 2018). Currently, though, automatic content moderation mainly focuses on a single utterance rather than on the ongoing conversation. Whether a message fosters agonism depends on various dimensions, including which groups are participating, how many participants, their level of interaction, and their relative power differences, all subject to

change over time.

We introduce a methodology for collecting and curating a dataset of English Twitter conversations embodying various aspects of dimensions of conflict. We collected conversations about trending events, as these often serve as catalysts, prompting individuals to interrogate their stance on current issues and deliberate their self-conception in relation to their views. We then annotated the conversations following our own coding schema, created with antagonistic and agonistic conflict in mind. These annotations took context into consideration rather than focusing on an isolated tweet.

Next, we trained logistic regression and transformer models (BERT and GPT-3) on this dataset, to predict dimensions of conflict. The models are trained on human annotations of the conversations, enhanced with a) previously proposed labels for online conversation: constructiveness and toxicity; and b) contextual aspects such as cardinality (participant counts) and topology (interaction structure). We show that all models are sensitive to the specific words in the conversation, making them less generalizable across topics and domains. Incorporating conversational context, however, makes the models more robust, showing that cardinality and topology are important dimensions in the prediction of online conflict.

The potential use cases of this work include the measurement of productive (agonistic) versus unproductive (antagonistic) conflicts, providing insights into where learning and constructive discourse can be fostered. This work ties in with attempts at reflective content moderation, where the goal is not simply to delete harmful content or make it less visible (Zeng and Kaye, 2022), but also to identify and promote content that can be constructive and productive towards democratic goals (Mouffe, 2013; Gillespie et al., 2020; Morrow et al., 2022). This kind of analysis has implications for detecting early signs of scapegoating and unproductive disputes—cases where patterns of discourse do not necessarily break the terms of service, but can nonetheless bring harm over time. Our findings will be instrumental in shaping online discourse, aiming to harness conflict as a driver of democratic conversation (agonism) rather than as a destructive silencing element (antagonism).

Our main contributions are: a) a conceptualization of conflict online on a scale between antagonism and agonism; b) a process to retrieve conflict-ual conversations on Twitter; c) a detailed schema to annotate conversations for various dimensions of conflict and the resulting annotations, with good inter-annotator agreement; and d) a set of experiments that show the usefulness of contextual information in predicting online conflict.

## 2 Related Work

### 2.1 Conflict and agonism

Within this project, we understand conflict as the generative ground that spans hate to agonism. According to political theorist Chantal Mouffe (2013), democratic speech or dialogue always bears traces of the conflicts from which it emerges. Democracy, therefore, necessarily entails conflict and negotiation; to expand who counts as a citizen, to negotiate differing claims to freedom or rights, and to validate collective decisions. This is agonistic conflict. Conflict can also be antagonistic: unproductive and undemocratic, when it silences individuals and groups, by shutting them out or harassing them with hateful speech. Democratic institutions are responsible for creating the space to allow conflicts to take an agonistic form, in which opponents are not enemies but adversaries among whom conflictual consensus may emerge (see also Mouffe, 2002; Rancière, 1999, 2010; Wenman, 2013). This generative aspect of conflict has been neglected in discussions around content moderation on social media platforms, which frame the problem as freedom of speech versus censorship (Douek, 2022; Gillespie et al., 2020; Gillespie, 2018).

### 2.2 Abusive language

Abusive language online is a broad term that covers various forms of harmful or offensive communication on the internet, such as hate speech, cyberbullying, trolling, or flaming (Fortuna et al., 2020; Pachinger et al., 2023). Detecting and preventing abusive language online is an important challenge for natural language processing (NLP) and social computing, and an extensive literature on the topic exists. In addition to the challenges of detecting a social phenomenon that the perpetrators often try to disguise, current industry solutions suffer from a lack of interpretability, undermining their credibility (MacAvaney et al., 2019). Equally dangerous is the over-policing of certain communities and topics online (Saleem et al., 2016).

Further, supervised classifiers require high-quality annotated data that may harm the annotators

and that may contain their biases (Sap et al., 2022; Vidgen and Derczynski, 2021). We know context is also crucial in obtaining high-quality annotations (Ljubešić et al., 2022), and that some disagreement among annotators is to be expected (Leonardelli et al., 2021). All this prior work informs our annotations and explorations of machine learning models.

We emphasize, however, that we do not necessarily correlate the absence of toxicity or abuse with the presence of productive conflict. Abusive language research tends to characterize healthy and/or civil conversations as those that are absent of toxicity (e.g., Smith et al., 2021; Hede et al., 2021). While that may be the case, healthy conversations are not necessarily agonistic. Agonism requires a certain level of disagreement as a source of political discussion and engagement.

## 2.3 Persuasion, argumentation, derailed conversations

Persuasion styles, rhetorical strategies, and argumentation styles all play a role in how we perceive and interpret conflict. Research in this area has produced manual annotations of rhetorical strategies such as framing, hedging, modality, repetition, and rhetorical questions (Peldszus, 2014; Green, 2014; Hirst et al., 2014). These approaches tend to focus on understanding which rhetorical approaches will be most effective in changing someone's mind (Habernal and Gurevych, 2016; Hidey and McKeown, 2018).

Modeling work includes attempts to find arguments in text, an area known as argumentation mining (Mochales and Moens, 2011; Lawrence and Reed, 2019; Harris and Di Marco, 2017). The goals include: a) identifying controversial topics in debates, news, Wikipedia articles, or online discussions (Boltužić and Šnajder, 2014; Kittur et al., 2007; Choi et al., 2010; Swanson et al., 2015; Stab and Gurevych, 2017); b) forecasting conversational derailment (Zhang et al., 2018); c) identifying conversational strategies that will change the interlocutor's mind as in r/ChangeMyView (Monti et al., 2022; Srinivasan et al., 2019); d) detecting conflict outside the conversation, as in r/AmITheAsshole posts (Welch et al., 2022).

While this previous research on persuasiveness informs ours, its goal is to identify successful and unsuccessful argumentation styles. We are, first, looking for conflict, to then try and pinpoint examples of agonistic discussions, which do not necessarily have a successful outcome in terms of persuading interlocutors.

## 2.4 Constructive comments

Research into high-quality online content has shown that constructiveness is a useful dimension. Constructive comments build on and contribute to the conversation, providing points of view and justification for a particular opinion. They are not necessarily conflictual in nature, since they may simply build on the ongoing conversation. In a study of online news comments, Kolhatkar et al. (2023) propose that constructive comments seek to create a civil dialogue, with remarks that are relevant to the article and not merely emotional provocations. Comments identified as constructive can be presented to future posters as prompts or examples of desirable behaviour or as nudges to depolarize conversations (Stray, 2022). Our work on identifying conflict can contribute to the growing body of research on how to present content in such a way that it contributes to productive, civil, and also agonistic discussion.

# 3 Dataset

## 3.1 Data collection

We are interested in online conversations on contentious topics, so we used the Twitter Academic API v2 elevated access to gather replies containing certain keywords, starting with controversial topics. Then we consulted subject experts and representatives of equity-seeking groups as a way to increase the topic diversity within the dataset. This led to a set of keywords as search terms (in Appendix A).

After selecting tweet replies in English containing the relevant keywords from each topic, we then extracted their surrounding conversation trees using two traversal methods: depth and breadth. The former involved recursively collecting a reply's referenced tweet until it reached the root message, or the 7-message limit (since length 7 is the last most frequent distribution before the start of the long tail of conversation thread). Breadth traversal involved capturing adjacent messages of a conversation by recursively creating new queries based on each reply's tagged author and the conversation ID of the reply.

The annotated dataset contains an equal mix of depth and breadth traversals. While the former enables more efficient data collection, the latter is useful for capturing the chaotic nature of conversa-

tions on most platforms, such as the one depicted in Figure 2.

Only conversations of length 3-7 messages in English were stored. This iterative process continued over a period of three years (January 2020 - December 2022), yielding a total of 220,626 conversations.[1]

Based on a random sample of 1,000 conversations, roughly 30% of these conversations likely involved first-time interactions[2], suggesting that these topics were contentious enough to spark debates among strangers in the comments of large accounts, creating virtual public forums.

## 3.2 Coding schema

To label the dataset, we developed an original coding protocol based on an interdisciplinary review of literature on conflict, including media studies, political science, conflict resolution studies, and critical race theory (e.g., Oetzel and Ting-Toomey, 2006; Lamberti and Richards, 2019; D'Errico et al., 2015; Itten, 2019; Yardi and boyd, 2010; Han et al., 2023).

The initial protocol was first tested on a subset of conversations (see Section 3.3). We revisited the coding protocol twice throughout the course of the project based on coders' feedback and discussions over disagreements. The final coding protocol followed a decision tree structure, where answering in the positive to one question led to a set of follow-up questions, as shown in Figure 1. Appendix E contains an extensive discussion of each of the concepts in the figure, with examples.

## 3.3 Annotation and agreement

We recruited a team of four annotators. In selecting candidates, we aimed to maximize demographic diversity and cultural background. The team included two graduate and two undergraduate students from three departments at our university: Communication, Political Science, and International Studies. The self-identified gender split was three women and one man, and the age ranges were: three 20-25 and one 26-30. At the beginning of the project, three members of the research team led a training

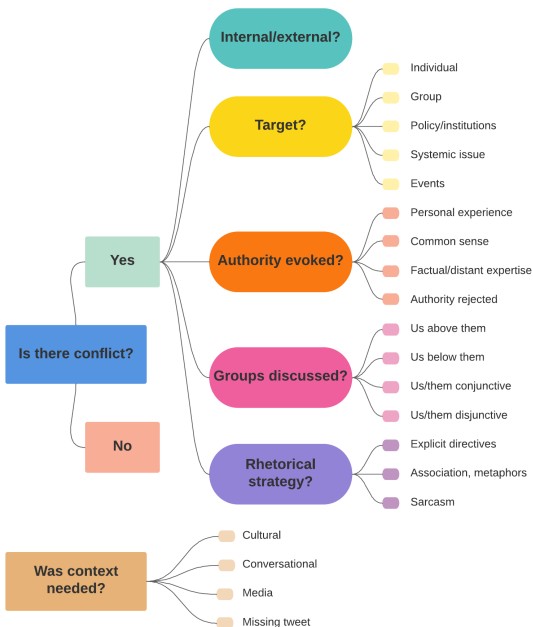

Figure 1: Coding schema for conflict

session during which annotators were introduced to the project and its goals and were taught how to use the labelling platform (LabelStudio; see Appendix E for screenshots). During the first week, the annotators and two members of the research team worked together and labelled a subset of about 400 tweets. The annotation was conducted in person, and each session had planned moments for discussing disagreements and clarifying the gray areas of the annotation protocol. At the end of the training session, we assigned a subset of the dataset to be codde. The research co-leaders held weekly individual check-in meetings with annotators to troubleshoot issues and gather their feedback. These meetings also served to assess the emotional impact of annotation and to externalize the thoughts and emotions annotators encountered during their work. The entire research team also met every two weeks, to compare different annotation styles and discuss edge cases as a way to test the protocol's reliability. Each message was annotated by two annotators.

Throughout the duration of the annotation campaign (May-December 2022), annotators labelled 4,022 conversations involving 9,472 individual authors with 22 labels. It was a two-tiered labelling system, where the annotators would first read the entire conversation and indicate whether there was conflict in the last message of the conversation given the context of the previous messages. They

---

[1]Raw data, annotations, and code used to extract the conversations are available in our repository, which also includes all the code for the experiments in Section 4: https://github.com/Digital-Democracies-Institute/Dimensions-of-Online-Conflict

[2]First-time interactions were approximated by examining each account's prior 200 messages and checking if any of the accounts had interacted with each other previously.

would then annotate further aspects if the answer was 'yes', following Figure 1.

Since each binary label was annotated by two annotators, we can compute inter-annotator agreement using Cohen's kappa, $\kappa$ (Cohen, 1960). The initial question, 'did the last message in the conversation contain conflict?' had $\kappa = 0.65$, a moderate to substantial level of agreement (Landis and Koch, 1977). The kappa values for the follow-up questions in Table 1 show that some of the other labels foster less consensus. Annotators agreed, in general, whether the conflict is internal or external to the tweet (with internal more difficult to adjudicate). A conflict is internal when all involved parties are also engaged in the conversation, i.e., it is conflict among the participants. An external conflict is disagreement about somebody else not in the conversation, e.g., a public figure (see Appendix E for more detail). Annotators also often agreed that conversational context was needed. The level of agreement for rhetorical strategies (sarcasm, explicit directives and calls to action, association and metaphor analogies) was quite low, although consistent with the well-documented difficulty in annotating sarcasm (González-Ibáñez et al., 2011; Oprea and Magdy, 2020). In this paper, we mainly use the binary label for conflict in the tweet, leaving other features for further study. We kept only conversations where the two annotators agreed on whether there was conflict. This yielded a total of 3,577 data points. One could argue that, by keeping only cases with clear agreement, we are in effect making the task 'easier'. Given the small size of the dataset, we follow this approach in order to eliminate noise.

Table 1: Cohen's kappa $\kappa$ values for different features

| Feature | $\kappa$ |
|---|---|
| Conflict (overall) | 0.65 |
| Target - Individual | 0.90 |
| External | 0.58 |
| Context - Conversational | 0.58 |
| Internal | 0.48 |
| Rhetorical strategy - Sarcasm | 0.36 |
| Rhetorical strategy - Explicit | 0.35 |
| Rhetorical strategy - Association | 0.19 |
| Context - Media | 0.18 |
| Context - Cultural | 0.02 |

Appendix B provides a trend line for level of activity per topic over time. We saw that some topics, like 'Social Distancing' were discussed over long periods of time, whereas other topics peaked and declined quickly, perhaps having to do with specific events ('Will Smith Slap', 'Rogers Out-

age'). Appendix C provides further information on the rate of conflict per topic. In summary, most topics contained some form of conflict, due to the way they were collected (trending topics). This makes the dataset possibly unbalanced, but also a rich source of how conflict develops online.

# 4 Conflict predictive models

For now, we focus only on finding conflict, leaving the issue of whether the conflict is agonistic or not for future work (although we make some suggestions in Section 5). Our starting hypothesis is that we can find signals of conflict in the data. The target variable of interest in this paper is the presence of conflict. Predictive features include: words in the text (bag of words model); constructiveness and toxicity labels; and context from the conversation, namely cardinality (number of participants) and topology (structure of participant interactions).

By conversational topology we refer to the multi-threaded nature of online conversations, which have been described as polylogues (Marcoccia, 2004). Let us examine it with an example, represented in Figure 2. Amal sends out a message about a new mural in their city. Boróka angrily replies that this mural is a waste of tax-payer money. Deniz replies to Boróka with a meme making fun of Amal, and Boróka sends the laugh emoji back to Deniz in response. Carlu tells Amal that a section of the mural is controversial and divisive and shouldn't have been publicly-funded. Eryl replies to Deniz clarifying that the section of the mural they are referring to was not actually funded by the city. Eryl then says the same thing to Carlu.[3]

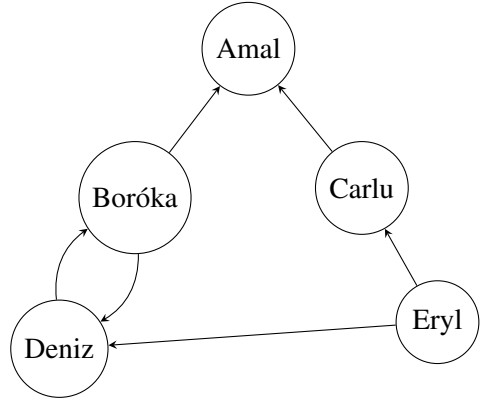

Figure 2: An example conversation represented as a graph

Each directed edge embeds information about

---

[3]Names from the LEDIR repository (Sanders et al., 2020).

who received a notification from whom. If there is a path from C to A, that means that A received a notification from C. For the above example, Deniz's message to Boróka sends a notification to both Amal and Boróka, but not to Carlu. Intuitively, it should help to know the directed graph's structure of a conversation. It seems important to know, for example, that an utterance is part of a larger conversation sending out a notification to five different people versus a back-and-forth conversation between just two people. For the experiments, we encoded this feature as a binary 'has_bidirectionality' feature (i.e., is there back-and-forth interaction), but the dataset we will release has a full representation of this dynamic. For instance, the conversation in Figure 2 is represented as: [(B, A), (D, B), (B, D), (C, A), (E, D), (E, C)], which is also converted to a matrix form in the annotated dataset.

## 4.1 Bag of Words model

The first basic hypothesis (**Hypothesis 1a**) is that the presence of some words (unigrams and bigrams) is predictive of conflict. We used a logistic regression classifier with L2 regularization to predict whether the entire conversation represented conflict or non-conflict. It used a TF-IDF vectorizer to extract unigram and bigram features from the last message in the conversation, the message that we first extracted using keywords. We also used not just the last message, but the entire conversational context, with the same model (**Hypothesis 1b**). When we ran this model, we saw that some terms had coefficients with very high absolute values in predicting the presence of conflict. Although some of these terms are intuitively indicative of general conflict, such as the unigram 'people', some of the terms are most likely hyper-specific for particular conflicts that will have only happened once, such as the unigram 'smith', referring to the Will Smith slapping incident. To reduce the risk of overfitting this model to particular topics on a new dataset or domain, and to have the model learn topic-agnostic linguistic patterns of conflict, we removed topic-related unigrams and bigrams by selecting the top 10 c-TF-IDF words from each topic and then removing those that seemed highly topic-specific (see Appendix D for lists of words removed by topic). We use this topic-agnostic dataset (Dataset 2) for the fine-tuning experiments in the next section. Dataset 1 contains all the words, without filtering.

We also postulate (**Hypothesis 1c**) that the bag-of-words approach can be enhanced with additional labels. These labels are derived from other models that classify messages according to their constructiveness (Kolhatkar et al., 2023) and toxicity (Hanu and Unitary Team, 2020).[4] We added logit scores from each of these existing models into the same feature matrix. Table 2 shows a confusion matrix for the bag-of-words (BOW) model with constructiveness and toxicity on the 716 conversations in our test set (we used a standard 80-20 split for training and testing). We can see that this model is quite good at identifying both non-conflict and conflict, but it overpredicts conflict, as it is the majority in the imbalanced dataset.

| | Non-Conflict | Conflict | Total |
|---|---|---|---|
| **Non-Conflict** | 27 | 87 | 114 |
| **Conflict** | 0 | 602 | 602 |
| **Total** | 27 | 689 | 716 |

Table 2: Confusion matrix for BOW LR model with constructiveness and toxicity (Hypothesis 1c)

## 4.2 Transformer model

Our second main hypothesis (**Hypothesis 2a**) is that we can detect conflict using a transformer model, the BERT (Devlin et al., 2019) implementation from HuggingFace.[5] Further, we propose that, beyond the specific words in the message, the context of the conversation contributes to its likelihood of becoming conflictual. We examine different types of contextual information: the previous messages (**Hypothesis 2b**); the previous messages with constructiveness and toxicity labels, as we saw in Section 4.1 (**Hypothesis 2c**); the cardinality of the conversation (**Hypothesis 2d**); and the topology, or structure of the conversation (**Hypothesis 2e**).

We use a technique inspired by Jin and Aletras (2021) to incorporate contextual information into the BERT model. This approach has proven effective for complaint severity classification by injecting linguistic features. The word representations from the embedding layer are combined with the contextual information using an attention gate to control the influence of different features. The combined representations are passed through the BERT encoder, followed by an output layer. We set the

---

[4]https://github.com/unitaryai/detoxify
[5]https://huggingface.co/bert-base-uncased

max length to 256 and keep the parameters in the attention gate the same as Jin and Aletras (2021).

## 4.3 Comparison of model results

Table 3 shows F1 score results for the two main approaches we took, following the hypotheses described above. We first show results for a BOW logistic regression model under the three different conditions (last message, all messages, all messages with constructiveness and toxicity). We then test the performance of BERT with the same conditions, plus we incorporate cardinality and topology. We also tested a simple GPT-3 fine-tuned model using OpenAI's Curie fine-tuning API. Our test data comes from the human annotations (Dataset 1), and the same data but with topic-related words removed (Dataset 2). The latter is more likely to be generalizable to new data about different topics, which is why we are interested in performance changes relative to Dataset 1. Results in the table are F1 score averages (average of three runs with random seeds 42, 43, 44).

| Model | Dataset 1 | Dataset 2 |
|---|---|---|
| LR Last Msg (H1a) | **38.30** | **33.82** |
| LR All Msg (H1b) | 26.87 | 23.73 |
| LR All Msg + constr, tox (H1c) | 38.30 | 24.81 |
| BERT Last Msg (H2a) | 94.58 | 85.26 |
| BERT All Msg (H2b) | **96.06** | 85.42 |
| BERT All Msg + constr, tox (H2c) | 95.43 | 87.26 |
| BERT All Msg + cardinality (H2d) | 92.83 | **89.36** |
| BERT All Msg + topology (H2e) | 94.40 | **88.92** |
| GPT-3 Fine-tune All Msg | 91.95 | 85.85 |

Table 3: F1 scores across datasets. constr = constructiveness labels; tox = toxicity labels

We can see from Table 3 that the simple logistic regression model (LR) results are quite low, but have a drop for Dataset 2 that is comparable to that of the BERT models. There are no gains whatsoever for the LR model from including more context, in the form of knowing the prior messages in the conversation, or more information, such as the constructiveness and toxicity labels. This might result from the absence of word sequence modelling in LR, which may hinder its ability to capture contextual dependencies between words.

The results from BERT are more interesting. First of all, a simple model with just the last message shows $F1 = 94.58$, with a $0.32$ drop if topic words are not present. Including all messages helps considerably, raising the score to its highest level, $96.06$, but with an ever larger drop for Dataset 2,

with no topic words. This improvement with all the messages likely results from BERT's ability to account for contextual sequences, as compared to the LR model.

Even more interesting is the effect of additional labels. When we incorporate constructiveness and toxicity, there is an improvement over the baseline of the last message and a slight decline from just including all the messages. However, the model is more robust to the removal of topic words. The models with cardinality and topology have similar topic-independent robustness. Cardinality leads to the lowest drop in performance ($3.47$ points) for Dataset 2, and topology also seems to show some topic independence. The GPT-3 model performs worse compared to all BERT combinations, and also suffers from a drop in Dataset 2. We should note that the GPT-3 model that we use was trained on data collected roughly up to the end of 2019 (Brown et al., 2020), so it lacks knowledge about most of the topics in our data, which was collected later. It serves, thus, as a good test for topic independence.

We conclude that information about the conversational context is useful in pinpointing conflict, additionally contributing the type of contextual information that the model needs to be robust to changes in topics and individual words.

## 5 Approximating agonism

We can attempt to approximate agonism as well as other categories of these conversations given the three dimensions shown in Figure 3 by defining $P_A$ as the Possibly Agonism Score, $P_U$ as the Possibly Unproductive Score, and $P_S$ as the Possibly Small-Talk Score as follows:

$$P_A = 1 - \sqrt{(T - 0.0)^2 + (S - 1.0)^2 + (C - 1.0)^2} \quad (1)$$

$$P_U = 1 - \sqrt{(T - 1.0)^2 + (S - 0.0)^2 + (C - 1.0)^2} \quad (2)$$

$$P_S = 1 - \sqrt{(T - 0.0)^2 + (S - 0.0)^2 + (C - 0.0)^2} \quad (3)$$

where $T$ is the Toxicity Likelihood, $S$ is the Constructiveness Likelihood, and $C$ is the Conflict Likelihood.

Using these proxy scores, we can compare the ratios of unproductive versus agonistic conversations across different trending topics over time, shown in Figure 3. Part of our future work involves a

qualitative analysis of all the conversations in that green zone, to investigate whether they have traces of agonism.

Furthering our analysis, we sampled the top 100 bidirectional conversations from the highest $P_U$ and $P_A$ scores. A different set of annotators was then tasked with categorizing each conversation as either agonistic, antagonistic, or neither. This resulted in an inter-annotator agreement quantified by $\kappa = 0.44$, indicating a moderate level of agreement. The distribution of resulting labels in agreement is shown in Table 4.

Table 4: Distribution of conversational labels

| Label | Percentage |
|---|---|
| Agonistic | 34% |
| Antagonistic | 32% |
| Neither | 35% |

76.5% of the conversations coded as Agonism were sampled from the top 100 $P_A$ set, and 80% of the conversations coded as Antagonism were sampled from the top 100 $P_U$ set. These findings suggest that while our proxy scores could do moderately well at discerning between agonistic and antagonistic conversations, there is still room for improvement. Implementing a secondary machine learning model on a larger dataset with labelled data from this secondary annotation exercise appears to be a promising next step in our goal towards modeling agonism.

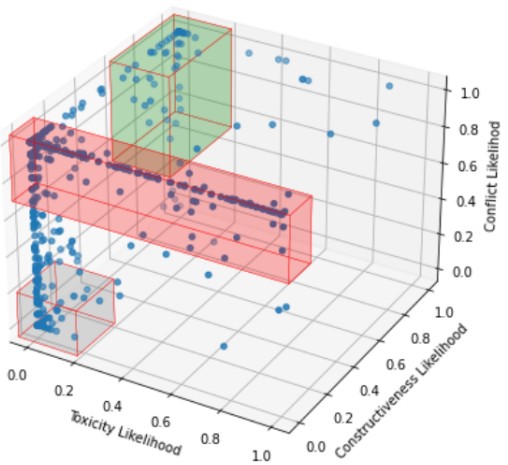

Figure 3: Conversations plotted by conflict, constructiveness, and toxicity likelihoods. We are postulating the green volume as the 'zone of agonism'.

## 6 Conclusions and future work

The long-term goal of our program of research is to identify agonistic conflict and distinguish it from less productive and democratic forms of conflict. The work presented here contributes a definition of agonism and its operationalization in a coding schema, an important step in approaching content moderation as a task of fostering agonistic dialogue, a productive form of conflict that is essential to democratic dissent.

We introduce a richly annotated dataset of online conversations containing conflict. Using this data, we test methods that can identify conflict from conversations, crucially incorporating contextual information. We experiment with dimensions of the context that we believe can be proxies for agonistic conflict, including the presence of constructiveness and toxicity, the number of participants, and the topology of the conversation, which includes the level and direction of interaction. We show that the contextual information is key to identifying conflict, especially because it helps the model remain topic-agnostic. This contextual approach can be helpful not just in identifying conflict and agonism, but also in detecting abusive language, as it provides a wider view of the conversation, rather than whether an utterance is abusive or not in isolation.

We have made data and code available in a repository.[6] This includes: the dataset of 4,022 conversations with annotations, the code to collect conversations, the LabelStudio annotation scheme, and the code for all the experiments described. The appendices in this paper include detailed information about the data collection process, the coding schema, and multiple examples of annotated conversations from the dataset. The repository also contains links to raw data, a larger dataset of conversations we collected but have not annotated (the entire dataset contains 220,626 conversations). We will, additionally, make available a demo web application linked in the repository to experiment with the model results.

The next steps in our research program involve deploying the other dimensions in the data (us vs. them conflict; individual/group conflict, etc.). We also plan to perform a qualitative analysis of conversations with high conflict, high constructiveness, and low toxicity, which we have defined as a potential zone of agonism. Further experiments will ex-

---

[6]https://github.com/Digital-Democracies-Institute/Dimensions-of-Online-Conflict

tend this model to other topics in our larger dataset, to test whether it can be used for semi-automatic annotation. We also plan to gather additional topics that generated discussion by querying keywords from monthly snapshots of the Wikipedia Portal:Current_events and the Top_25_Report through the Wayback Machine.

Defining and identifying agonism in conversations is a difficult task. We also acknowledge the difficulty of fostering the kinds of spaces that are conducive to agonistic debate, both online and offline, which is why an interdisciplinary approach with both quantitative and qualitative approaches such as the one presented here is needed.

## Limitations

Research on conflict consistently draws attention to its complex nature. Peace studies scholar Giorgio Gallo (2013) states that conflict can be characterized by multiple, diverse, at times hidden, undefined, and evolving objectives. Thus, translating such a complex phenomenon into a set of labels output by a machine learning system is reductionist at best. Gallo notes that most research on conflict tends to isolate it from its context, thus oversimplifying the inquiry. We note that ours is one such simplification. The nature of online conflict, with long conversations unbounded by time and space limitations, unlike face-to-face communication and debate, renders more fine-grained and contextualized approaches impossible. We nevertheless attempt to incorporate context beyond the words in the individual messages or message threads, by examining features of the conversation, the number of participants, and the topology of the conversation.

Precisely because there is so much conflict online, a dataset with about 4,000 conversations is not a representative sample. The method of collection, where we started with topics likely to generate conflict (both hateful and agonistic), may also result in biased data. One alternative we could contemplate is to draw from datasets that have been labelled for toxicity, as those are more likely to contain conflict. Such data, however, does not necessarily contain agonistic conflict. Furthermore, as we mention in the conclusion section, we have not yet reached the stage of identifying agonism automatically. We hope that, by detecting conflict overall, we can extract many instances automatically, leading us to a method for distinguishing antagonistic conflict from agonism. The main limitation of our study is

that we are at the early stages of a long and complex research process.

Beyond identifying individual comments as constructive, productive, or leading to agonism, it is also important to acknowledge the role of user and interface design in how comments are produced and presented (Masullo et al., 2022).

We also note more common limitations, including the source of the data (Twitter), the language of study (English), and the lack of demographic information about the participants in the discussions. We do not include information about language varieties and rhetorical strategies that may be characteristic of some online groups and linguistic communities and not in the mainstream. We do not know whether the conversations are representative of some mythical mainstream culture or of demographic groups with their own norms of debate and argumentation.

## Ethics Statement

We adhere to the ACL Ethics Policy. In particular, we strive to contribute to societal well-being and the public good by studying how conflict evolves in online conversations. We take the directive to avoid harms quite seriously. To avoid individual harm and respect privacy, we anonymize the tweets before releasing them publicly (although the Twitter ids will provide a link to the original). We also take into account duty of care for our researchers and annotators, and have provided them with opportunities to debrief and protect their mental health when they read conflictual and hateful material.

We are concerned about the damage to the environment caused by training and fine-tuning large language models. We mitigated, in part, by using only pre-trained models. The BERT model was fine-tuned on a data center powered by hydroelectric energy, thus producing fewer $CO_2$ emissions.

Although our current system is far from perfect in detecting conflict, and we have not yet produced a method to detect agonistic conflict, one risk of such systems lies in their misuse by employers, governments, or other social agents to quell agonistic and productive conversations. For example, an employer may wish to suppress agonistic discourse that could lead to employee unionization.

## Acknowledgements

We express our deepest gratitude to the coding team members who contributed to the success of this

project: Pranjali Jatinderjit Mann, Ashley Currie, Umer Hussain, and Judy Yae Young Kim. Additionally, we are thankful for Zeerak Talat's assistance in developing and launching the project.

This research was supported by the joint contribution of the UK Economic and Social Research Council and the Canadian Social Science and Humanities Research Council. The study was conducted as part of the "Responsible AI for Inclusive, Democratic Societies: A cross-disciplinary approach to detecting and countering abusive language online" project [grant number R/163157-11-1].

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

## A  Appendix: Search keywords

The following are the list of keywords used for the search described in Section 3.1.

(1) #billc11, #cancon, #CRTC, #onlinecensorship, #onlineharmsreduction, #StopAsianHate, #stopC11, bean dad, bell hooks, biden loans, bill gates divorce, canadian truckers, capitol insurrection, china gamer ban, coastal gas link, convoy, defund the police, depp heard, el salvador bitcoin, elon twitter, evergrande, gamestop, groomers, hbo test, india pakistan missile, ivermectin, libsoftiktok, metaverse, ocean fire, online harms, pandora papers, realdonaldtrump, rittenhouse, robb elementary, roe, rogers, sci_hub, social distancing, suez canal, taliban, tigray, tmx pipeline, west elm caleb, will smith slap, vaccine

## B    Appendix: Topic distribution over time

Figure 4 shows the number of conversations related to each topic over time. We can see that certain topics exhibit a more enduring presence in the discourse (*Social Distancing*), while others appear to be more transient, capturing attention for only a day or two (*Will Smith Slap, Rogers Outage*). Each topic was sampled from its peak of discussion. We have observed that topics with high peaks (*Will Smith Slap* or *India Pakistan Missile*) tend to have a higher level of conflict than topics that last over longer periods of time.

## C    Appendix: Summary statistics for top topics

Table 5 presents summary statistics for the top topics (a total of 2,718 conversations, out of the 4,022 in the dataset). The first column shows the total number of conversations in that topic. The next few columns display the percentage of conversations in that topic with the label (e.g., 62% of the Social Distancing conversations contained conflict, and 11% had sarcasm). We see that most topics contained some form of conflict (making the dataset unbalanced), but that presence of other dimensions of conflict varies widely. Using the percentage of conflict per topic in Table 5, we can test for statistically significant differences in conflict rates across topics. Assuming non-normality and non-equal variances, we can use the Kruskal-Wallis H test defined as follows:

$$H = \frac{12}{N(N+1)} \sum_j \frac{R_j^2}{n_j} - 3(N+1) \quad (4)$$

This calculation obtained a p-value $< .005$. We can confidently say that topics are clearly different in their rate of conflict, with discussions about *Biden's loan forgiveness program* being 100% conflictual, and conversations about *Facebook's Metaverse* being low in conflict (8% of the conversations in that topic had conflict).

## D    Appendix: Filtering out keywords

The following is the list of keywords removed from each topic as described in Section 3.1.

- Jan-6th-insurrection: capitol, insurrection, trump
- TMX-pipeline/coastal-gaslink: pipeline, tmx, trudeau
- biden-loans: loans, student, biden
- canada-day: canada
- canadian-truckers: truckers, canadian, convoy
- canadian-truckers, protests: truckers, protests, canadian, canada

- defund-the-police: police, defund
- defund-the-police-realdonaldtrump: police, realdonaldtrump, defund
- depp-heard: depp, heard, amber, johnny
- elon-twitter: twitter, elon, musk, elonmusk
- groomer: govrondesantis, groomers, travlingsnowman
- india-pakistan-missile: pakistan, india, indian
- libsoftiktok: libsoftiktok, taylorlorenz
- metaverse: metaverse, crypto, bsc
- protests-social-distancing: protests, distancing
- queen-elizabeth: us
- roe-v-wade: roe, abortion, wade
- russia-ukraine: ukraine, russia
- salman-rushdie-attack: rushdie, salman
- social-distancing: distancing
- tigray-ethiopia: tigray, ethiopia, tplf
- vaccine: vaccine, covid

## E    Appendix: Coding schema

**Content warning:** Examples in this section contain offensive language.

Annotators were presented with a LabelStudio interface that provided: the tweet to annotate, the previous 3-7 messages in the conversation, and the labels to choose from, as seen in Figure 5. Labels were always for the last tweet in the conversation, but in the context of the entire thread. Here, we elaborate on the descriptions for each annotation decision, from the schema in Figure 1.

### E.1    Is there conflict?

A yes/no answer, based on the definition of conflict provided (see Section 2.1). Annotators were instructed to label only the last message (e.g., the last tweet in Figure 5), but use the context (the previous tweets) if necessary. A 'yes' answer triggers all the decisions below.

### E.2    Internal/external

If the annotators decided that there was conflict, they had to label the conflict as internal to the conversation ('when it involves people/entities directly engaged in the conversation') or external. For example, if people are discussing Black Lives Matter as an organization, then the conflict is external. But if they discuss people in the conversation involved in BLM, then it's internal. A tweet can be both internal and external, so both labels are allowed. This label had a relatively high level of agreement (see Table 1), so we feel this was a valid distinction. Examples (2) and (3) show instances of each.

(2)  `[Internal]` You are going to arrst me for standing 4' away from someone? Go for it

(3)  `[External]` Honestly at this point second wave is gonna have to happen for it to get through their THICK SKULLS that they completely and utterly fucked this up

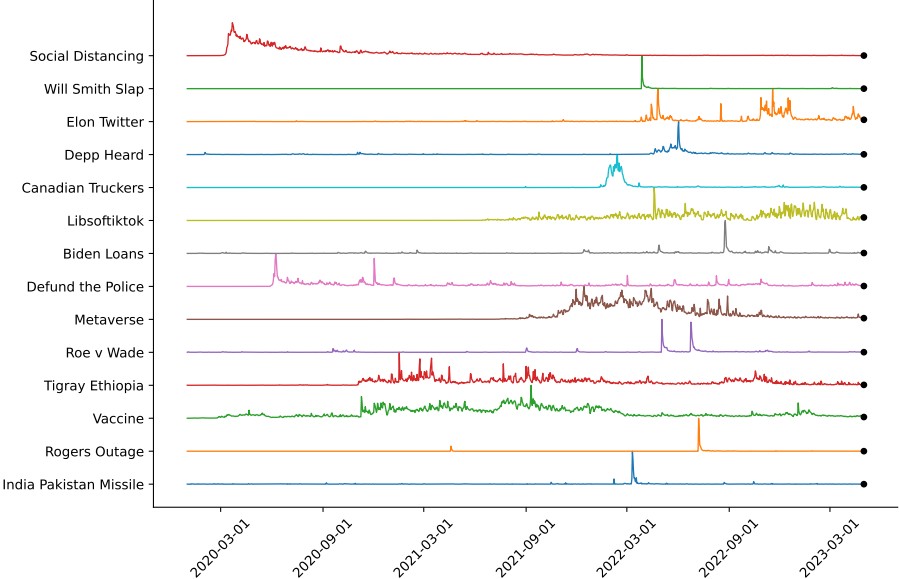

Figure 4: Volume of messages for some of the top topics over time

| Topic | N | Conflict | Sarcasm | Metaphors | Directives | Target |
|---|---|---|---|---|---|---|
| Social Distancing | 615 | 0.62 | 0.11 | 0.36 | 0.29 | 0.40 |
| Will Smith Slap | 320 | 0.97 | 0.20 | 0.76 | 0.05 | 0.96 |
| Elon Twitter | 268 | 0.90 | 0.23 | 0.67 | 0.14 | 0.91 |
| Depp Heard | 233 | 0.97 | 0.11 | 0.70 | 0.08 | 0.95 |
| Canadian Truckers | 230 | 0.97 | 0.10 | 0.67 | 0.15 | 0.53 |
| Libsoftiktok | 191 | 0.97 | 0.18 | 0.55 | 0.09 | 0.80 |
| Biden Loans | 190 | 1.00 | 0.13 | 0.72 | 0.23 | 0.78 |
| Defund the Police | 157 | 0.94 | 0.06 | 0.41 | 0.42 | 0.33 |
| Metaverse | 131 | 0.08 | 0.18 | 0.50 | 0.05 | 0.59 |
| Roe v Wade | 91 | 0.95 | 0.10 | 0.38 | 0.19 | 0.43 |
| Tigray Ethiopia | 90 | 1.00 | 0.06 | 0.68 | 0.40 | 0.31 |
| Vaccine | 76 | 0.89 | 0.20 | 0.24 | 0.30 | 0.45 |
| Rogers Outage | 69 | 0.48 | 0.15 | 0.20 | 0.30 | 0.41 |
| India Pakistan Missile | 57 | 1.00 | 0.21 | 0.61 | 0.12 | 0.46 |
| Total/average | 2,718 | 0.84 | 0.12 | 0.49 | 0.16 | 0.56 |

Table 5: Top topics and percentage of conversations in those topics that were given select annotation labels

### E.3 Who/what is the target?

The target of the conflict can be more than one, so multiple labels are possible here. Some examples are presented in (4)-(6).

- Individual
- Group
- Policy/institutions
- Systemic issue or cause
- Events
- Other (specify)

(4) [Individual] Cummings should be sacked! And prosecuted in criminal law.

(5) [Group] Its unreasonable and unrationale to say that we can't let emotions take over. Imagine being a black person in America right now. We should be angry, we shouldn't be pushing down or hiding our feelings. We want justice, everything else is not important right now.

(6) [Policy/institutions] That assumption is based on a busy car park. With miles of beach I am sure it's easy to stay far enough away from others. Visiting Tesco's is 100x more dangerous

### E.4 Authority evoked

This question asked whether the position, stance, or claim is recognized, consistent with, legitimated, or supported by some form of authority. Annotators were instructed to consider how different forms of authority are invoked by the participants, again focusing on the last tweet in the conversation.

- Personal experience. This could be an individual encounter, an understanding, or an insight, usually derived from proximity and epistemic positions.

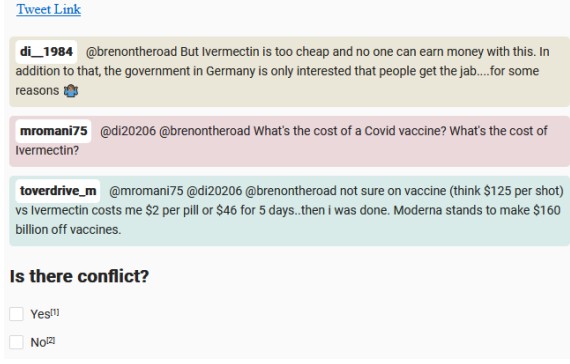

Figure 5: Annotation interface

- **Common sense.** Authority is evoked by referring to common sense knowledge, some sense of what is shared in the specific or cultural context (regardless whether common sense is actually true).
- **Factual/distant expertise or institution.** Participants support their statements by referring to experts, facts, or (statistics provided by) institutions.
- **Authority is rejected.** Participants reject or challenge authorities invoked by others.
- **Other (specify)**

(7) [Personal experience] Pretty sure for my son to go to boarding school he needed proof of vaccinations

(8) [Common sense] The mothers who raise these men play a role. They, too, are responsible. Many of them support the Taliban. It is naive to believe otherwise

(9) [Factual/distant expertise or institution] All indications are that the practice of sex with social distancing will have a profound effect on population control. Perhaps even larger that what 'the pill' had in its day when it first came out.

## E.5 Groups being discussed

This is a binary question, asking whether the conversation involves clearly identified groups or factions, presented as an attempt to render conflict as isolated incompatibility between groups. If the answer is 'yes', this triggers another set of questions, about the relationships among those groups and how they are being discussed:

- Us above them
- Us below them
- Us/them conjunctive. The two or more groups being discussed or involved in the conversation are presented as being allied, connected, or somehow related.
- Us/them disjunctive. The two or more groups are presented as not allied, connected, or related.

This label had a very low level of agreement with a kappa value of 0.02. We believe the explanation of this distinction, as shown above, was unclear to annotators, perhaps also because these oppositional relations are not often explicitly stated. Some examples are provided below. Note the dual label in (10).

(10) [Us above them], [Us/them disjunctive] idk why y'all continue to defend people who prove themselves over and over again the rich will always find ways to hide their money and avoid paying their dues, if we continue to let them

(11) [Us/them disjunctive] Please sign our petition to fire RCMP Commissioner Brenda Lucki here, and maybe we can finally get some accountability for an organization that has gone completely off the rails. Keep fighting for what is right,

## E.6 Rhetorical strategy

A 'yes' answer here means that the annotator saw persuasion or appeals to sensibilities and meanings that were grounded in linguistic techniques, language moves, or other linguistic mechanisms. If the answer was 'yes', then they were asked to specify what type of strategy was deployed:

- Explicit directives and calls to action
- Associations, metaphors, or analogies
- Sarcasm
- Other (specify)

(12) [Explicit directives], [Sarcasm] Yo ACLU and Amnesty International, calm down. Why sudden panic about "free speech" when Elon wanna buy Twitter?

(13) [Associations, metaphors, or analogies] So seems like evergrande stroke a deal which is good for now (kicking can down the road) and news just came out of China about reducing coal. Now just the Fed gotta worry about. Besides the fact that charts are looking ugly.

(14) [Sarcasm] So doxxing people who disagree with you is OK, but not doxxing people who agree with you. Got it.

## E.7 Meta questions

Finally, regardless whether the answer to the conflict question was yes or no, annotators were asked three further questions, about context and about their own reaction.

The first question was whether more context was needed, beyond the thread. As we see in Figure 5, annotators were asked to label the last message and were allowed to read the previous messages on the screen. But they also had the possibility to click on the tweet and look at the context on

the Twitter platform, including any media. If they clicked, they were asked to answer 'yes' to this question. Further, they could specify what kind of context was needed:

- Cultural. The annotator needed to know more about the issue at stake. This could be current news topics, subcultural trends online, or other aspects of the topic.
- Conversational. The conversation was missing some elements, perhaps earlier than the messages included in the annotation platform, which made it difficult to interpret.
- Missing content. Tweets in the thread had been deleted.
- Media. Videos or images in the thread were not available, but seemingly necessary for interpretation.

The second meta question asked about the emotional reaction of the annotator, simply asking 'How did you feel when reading the conversation?' Some suggestions were provided:

- Shock
- Sadness
- Disgust
- Anger
- Fear
- Confusion
- Indifference
- Entertained
- Hopeful

The third and final question was about level of confidence ('How confident are you about your analysis?'). Annotators were provided a 1-5 scale, with 1 being not confident at all (more context was needed, tweets had been removed, tweet was indecipherable), and 5 being a high level of confidence or that the interpretation was straightforward.

### E.8 Full annotation

Figure 6 displays a message (last message) and its context, to show the richness and complexity of the annotation scheme. Labels for the annotation of this example are given in Figure 7.

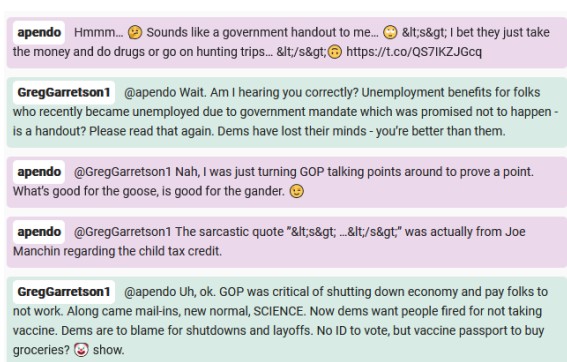

Figure 6: Conversation, with labels shown in Figure 7

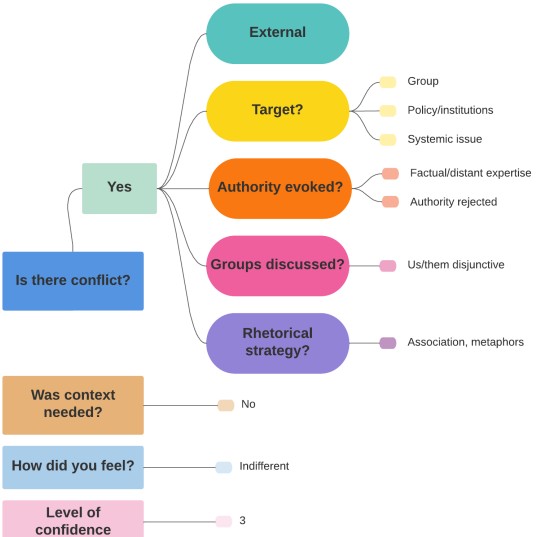

Figure 7: Full annotation for conversation in Figure 6