# OpenReview forum: "Dimensions of Online Conflict: Towards Modeling Agonism"
_EMNLP/2023/Conference — EMNLP 2023 Findings_

### Official Review · Reviewer_KsvZ · 2023-08-04

**Soundness:** 4

**Excitement:**

4: Strong: This paper deepens the understanding of some phenomenon or lowers the barriers to an existing research direction.

**Paper Topic And Main Contributions:**

This paper presents work on identifying the presence of conflict within Twitter data as an early step toward automatically distinguishing between productive, good faith conflict and conflict that is unproductive, hateful, etc. The primary contribution of this paper appears to be the creation of a dataset with useful human annotations, some early results regarding prediction of conflict, and some ideas for distinguishing between agonism and antagonism in future work.

**Reasons To Accept:**

1) The motivation to distinguish between productive and unproductive conflict will be of wide interest.
2) Robust data annotation processes were used.
3) Potential limitations are clearly highlighted.


**Reasons To Reject:**

 The first few sections of the paper seem to indicate that distinguishing between agonism and antagonism is the goal of the present paper. So it is a little disappointing to get to section 4 and realize that this paper is an early, somewhat speculative step toward that end. This could be cushioned a bit by making this clearer in sections 1-3.

**Reproducibility:**

4: Could mostly reproduce the results, but there may be some variation because of sample variance or minor variations in their interpretation of the protocol or method.

**Reviewer Confidence:**

5: Positive that my evaluation is correct. I read the paper very carefully and I am very familiar with related work.

---

> ### Author Rebuttal · Authors · 2023-08-29
>
> Thank you for a clear and encouraging review. We agree that it may be misleading to spend time in sections 1-3 talking about agonism, when we do not fully address the concept with our current findings. However, we believe it is important and valuable to establish the theoretical foundations of the work in conflict theory and communication. To clarify that this is the first step towards a longer-term project, we will reorient the introduction, while keeping the important explanations about the distinction that are of value to the current findings.
>
> As we mention in response to Reviewer APKa04, we are also in the process of enhancing the dataset, which includes our creation of a new data collection platform for Twitter/X, given the recent changes in access to their APIs. We will add an explanation of this new method/process to the paper if accepted.
>
> In our other responses, we also clarify how we will improve readability and expand on the motivation for the research project.

---

### Official Review · Reviewer_bZMt · 2023-08-04

**Soundness:** 2

**Excitement:**

3: Ambivalent: It has merits (e.g., it reports state-of-the-art results, the idea is nice), but there are key weaknesses (e.g., it describes incremental work), and it can significantly benefit from another round of revision. However, I won't object to accepting it if my co-reviewers champion it.

**Paper Topic And Main Contributions:**

This paper investigates modeling Agonism using Twitter data. The paper is motivated by the role of agonism in demoncratic dialogs and the need of better understanding on the difference of agonism vs hate. The authors contribute a dataset with a methodology for conflict annotation. Using the data, the authors conduct a set of experiments to verify their hypothesis. The results show contextual information is crucial for conflict identification.

**Questions For The Authors:**

I would like the authors to elaborate more on my doubts regarding the gap between motivations and experiments.

**Reasons To Accept:**

The paper is well-organized and easy to follow.
The paper contains a good amount of discussion on the role of conflicts played in online discussion. Studying the problem of differentiating whether conflicts are agonistic or hateful is very interesting.
The authors contribute a dataset for conflict study with detailed description of their methodology and annotation schema.


**Reasons To Reject:**

The paper is motivated for better understanding the natural of conflicts with discussion on the multiple roles of conflicts, e.g., fostering perspective or producing hate. However, the data and experiments are mostly about a binary prediction task on conflicts. I think there's a gap between the motivation and the experiments.

**Reproducibility:**

5: Could easily reproduce the results.

**Reviewer Confidence:**

4: Quite sure. I tried to check the important points carefully. It's unlikely, though conceivable, that I missed something that should affect my ratings.

**Typos Grammar Style And Presentation Improvements:**

The hypothesis is hard to track. It would be better if the authors can add a list of all the hypothesis.
Figure 1 and 3 are blurry.
The text in Figure 1 is hard to read.

---

> ### Author Rebuttal · Authors · 2023-08-29
>
> Thank you for the comments about the organization and clarity of the paper.
>
> Although it is true that there is a difference between predicting conflict and having a better understanding of the causes of the conflict, we've argued that establishing a baseline for being able to successfully detect conflict is a necessary first step in the direction of understanding the nature of conflict. The paper lays out the overall problem and contributes to this understanding with the dataset and an initial set of experiments. As reviewer KsvZ points out, the data annotation followed a robust process that we believe is an important contribution to the field.
>
> As we have discussed in our response to Reviewer APKa04, we will make the motivation more explicit and explain why the paper does not quite get (yet) to fully modelling agonism, because this is the first in a series of steps. Further, we felt that we needed a lot of space to document the problem and the dataset. We hope that this dataset and the lessons learned from our methodology can be used to help others working on a similar goal, and believe this is the kind of work that fits well in EMNLP, because it can advance the field by providing theoretical grounding and gold data for future experiments.
>
> We will re-draw the figures to improve readability.
>
> Finally, could you elaborate on what the technical or methodological problems are specifically? We are also not sure how to make the hypotheses easier to track. They are described in Sections 4.1 and 4.2, each in bold font, with corresponding results in Table 3.

---

### Official Review · Reviewer_APKa · 2023-08-05

**Typos Grammar Style And Presentation Improvements:** Figures should be modified to be tran…
**Soundness:** 3

**Excitement:**

4: Strong: This paper deepens the understanding of some phenomenon or lowers the barriers to an existing research direction.

**Paper Topic And Main Contributions:**

The paper focuses on detecting and differentiating two types of conflict to aid in platform moderation and content monetization. Twitter conversations related to trending controversial topics were collected, and a annotation schema was introduced to label various conflict dimensions, including the source, target, and rhetorical strategies employed. Logistic regression and transformer-based models, incorporating conversation context, were trained on the annotated dataset.

**Reasons To Accept:**

The introduction provides a clear overview of the paper's subject matter, introducing the concept of conflict online and its significance in democratic dialogue. The distinction between agonism and antagonism is well-defined, setting the foundation for the subsequent research.


The writing style is clear and concise, making the paper easy to follow.



**Reasons To Reject:**

The level of inter-annotator agreement for crucial aspects of conflict annotation is relatively low. While a moderate to substantial agreement was achieved for the binary conflict label (i.e., presence or absence of conflict), the lower agreement for other features, such as target, context, and rhetorical strategies, raises concerns about the reliability and consistency of the annotations.

The paper does not provide sufficient analysis or discussion regarding the reasons for the lower agreement on specific features.

To address this issue, the paper could explore potential strategies for improving inter-annotator agreement. This might include refining the annotation guidelines, or conducting additional training sessions to enhance consistency.

**Reproducibility:**

5: Could easily reproduce the results.

**Reviewer Confidence:**

5: Positive that my evaluation is correct. I read the paper very carefully and I am very familiar with related work.

---

> ### Author Rebuttal · Authors · 2023-08-29
>
> Thank you for the concise summary of the paper’s contributions and the appreciation for how well the problem was defined.
>
> We agree that agreement for some of the annotations is low. We’d like to point out that the last two lines in Table 1 are meta-questions, that is, not true annotation labels. Those meta-questions, "Context - Media" and "Context - Cultural" are not meant to be included as target labels for model training, but rather serve to gauge the annotator's opinion about the need for context to understand a given conflict. This could have been explained more clearly in the paper, so we will correct this in the revisions.
>
> Nevertheless, some of the other annotation labels have a relatively low level of agreement. This highlights the difficulty of the task for humans, which will likely make it a very difficult task for any NLP system. We believe the reasons for the low agreement in some of the categories are twofold:
>
> 1. Complexity of the task. The more abstract aspects, such as "Groups discussed", are challenging to discern with the limited context available in online conversations. Rhetorical strategies are notoriously difficult in annotation, especially sarcasm. This is well-documented in existing literature and it is a research problem in its own right (Sanguinetti et al., 2018; Castro et al., 2019).
>
> 2. Instructions and training. Annotators were told to focus on the conflict and the main aspects of the labelling (internal/external; target, etc.), and the more complex aspects may have received less attention. Joshi et al. (2019) report a wide range of inter-annotator agreement in sarcasm detection and suggest that framing in the guidelines is quite important. Our guidelines focus on the annotation of conflict and the high-level aspects. The annotation of rhetorical strategies received less attention and training.
>
> In any case, we only use the “conflict” label for the experiments, and “conflict” has a moderate agreement level. The main aspects of the labelling also have a moderate level of agreement. It's only when we reach the third-level classification (the terminal nodes in Figure 1) that the agreement decreases significantly.
>
> Ideally, we would have extended the annotation with clearer instructions and more data, as suggested by the reviewer. At the moment, we are in a difficult transition time, as Twitter/X changes its policies for research access and researchers are re-evaluating how much more data can be obtained from that platform.
>
> To address this issue, in the last month, we have developed a data collection platform that allows an individual researcher to save their Twitter/X reading data, as they scroll. We also plan to make the data and annotations not only available to all, but also editable. Users of the dataset can add their own annotations and comments, thus enlarging the data and contributing to higher inter-annotator agreement. The new annotations will be clearly marked to differentiate what is presented in this paper as a stable dataset from additions by the community.
>
> We will use the extra page in the final version to describe the new data collection platform and the process for adding annotations.
>
> We nonetheless believe this dataset, in its current format as used in the submitted version of the paper, is both rich and includes reliable labels for the top categories.
>
> We are planning a follow-up study to perform qualitative analyses of the data, which will include refining the labelling and further work on identifying agonism. If this paper is accepted into EMNLP, it will form the theoretical basis and main dataset from which to launch future work on identifying agonism vs. antagonism.
>
> Thank you for pointing out the need for more clarity in the figures. We will re-draw them to improve readability.
>
> **References:**
>
> Castro, S., D. Hazarika, V. Pérez-Rosas, R. Zimmermann, R. Mihalcea, and S. Poria. 2019. Towards Multimodal Sarcasm Detection (An _Obviously_ Perfect Paper). In Proceedings of the 57th Annual Meeting of the Association for Computational Linguistics, pages 4619–4629, Florence, Italy. Association for Computational Linguistics.
>
> Joshi, A., P. Bhattacharyya, and M. Carman. 2017. Automatic Sarcasm Detection: A Survey. ACM Comput. Surv. 50, 5, Article 73 (September 2018), 22 pages. https://doi.org/10.1145/3124420
>
> Sanguinetti, M., F. Poletto, C. Bosco, V. Patti, and M. Stranisci. 2018. An Italian Twitter Corpus of Hate Speech against Immigrants. In Proceedings of the Eleventh International Conference on Language Resources and Evaluation (LREC 2018), Miyazaki, Japan. European Language Resources Association (ELRA).

---

### Meta-Review · Area_Chair_oDHz · 2023-09-18

**Recommendation:** 4

**Metareview:**

Two  (APKa, KsvZ) reviewers feel strong excitement about this paper and find the soundness of the study to be quite sufficient in supporting its major claims. Reviewer bZMt urged authors to better clarify the paper’s seeming disconnect between the motivation and the operationalized tasks. Overall, there are some few issues that can be better addressed in the final version of the manuscript as suggested by the authors 1. Clarifying theoretical motivation in intro; tying back findings to theoretical motivation / terminology/ definitions (e.g., agonism), 2. Addressing low annotator agreement, 3. Making the motivation more explicit in connection to the tasks.

The authors address the reviewer questions/ concerns quite sufficiently in their rebuttal.

---

### Decision · Program_Chairs · 2023-10-07

**Decision:**

Accept-Findings

**Comment:**

Two  (APKa, KsvZ) reviewers feel strong excitement about this paper and find the soundness of the study to be quite sufficient in supporting its major claims. Reviewer bZMt urged authors to better clarify the paper’s seeming disconnect between the motivation and the operationalized tasks. Overall, there are some few issues that can be better addressed in the final version of the manuscript as suggested by the authors 1. Clarifying theoretical motivation in intro; tying back findings to theoretical motivation / terminology/ definitions (e.g., agonism), 2. Addressing low annotator agreement, 3. Making the motivation more explicit in connection to the tasks.

The authors address the reviewer questions/ concerns quite sufficiently in their rebuttal.